# A4C: Anticipatory Asynchronous Advantage Actor-Critic

**Tharun Medini** *
Electrical and Computer Engineering
Rice University, Houston, TX, USA
{tharun.medini}@rice.edu

**Xun Luan**\* **& Anshumali Shrivastava**
Department of Computer Science
Rice University, Houston, TX, USA
{xun.luan,anshumali}@rice.edu

## ABSTRACT

We propose to extend existing deep reinforcement learning (Deep RL) algorithms by allowing them to additionally choose sequences of actions as a part of their policy. This modification forces the network to anticipate the reward of action sequences, which, as we show, improves the exploration leading to better convergence. We propose a method that squeezes more gradients from the same number of episodes and thereby achieves higher scores and converges faster. Our proposal is simple, flexible, and can be easily incorporated into any Deep RL framework. We show the power of our scheme by consistently outperforming the state-of-the-art GA3C algorithm on popular Atari Games.

## 1 INTRODUCTION

There is a fundamental barrier in current Deep Reinforcement Learning(RL) algorithms, which is slow progress due to poor exploration. During the early phases, when the network is just initialized, the policy is nearly random. Finding a good action sequence via network exploration can take a significantly long time, especially when there are only very rare sequences of actions which give high rewards, while most others give low or zero rewards.

The current state-of-the-art algorithm for RL is Asynchronous Advantage Actor-Critic (A3C) (Mnih et al., 2016; 2015) which is not very conducive to GPUs. A follow up to this simple parallel gradient descent based algorithm was proposed by (Babaeizadeh et al., 2016) enabling the usage of GPUs (thereby called GA3C). It is the most optimized algorithm for RL particularly on Atari games, tailored for HPC platforms such as GPUs. Outperforming GA3C in running time is hard as it will require both algorithmic and systems advancement.

We present a simple opportunity of improving the convergence of deep RL and beat GA3C in time. In particular, we show that instead of learning to map the reward over a basic action space $\mathcal{A}$ for each state, we should force the network to *anticipate* the rewards over an enlarged action space $\mathcal{A}^+ = \bigcup_{k=1}^{K} \mathcal{A}^k$ which contains sequential actions like $(a_1, a_2, ..., a_k)$. While using the same episode information like A3C, we extract more gradient updates by training with action tuples apart from the basic actions.

## 2 OUR PROPOSAL: A4C

At a high level, our proposal extends the basic action set $\mathcal{A}$ to an enlarged action space $\mathcal{A}^+ = \bigcup_{k=1}^{K} \mathcal{A}^k$, which also includes sequences of actions up to length $K$. We illustrate our proposal in figure 1. Let us say $\mathcal{A} = \{L, R\}$ and we allow 2-step anticipation, therefore our new action space is $\mathcal{A}^+ = \mathcal{A} \cup \mathcal{A}^2 = \{L, R, LL, LR, RL, RR\}$. Each element $a^+$ belonging to $\mathcal{A}^+$ is called a meta-action, which could be a single basic action or a sequence of actions. Typical Deep RL algorithms have a DNN to output the estimated Q values (expected cumulative future reward like DQN (Mnih et al., 2013)) or policy distributions according to basic action set $\mathcal{A}$. In our algorithm, we instead let the DNN output values for each meta-action in the enlarged action set $\mathcal{A}^+$. If we see

---

*Equal Contribution of authors

each gradient update as a training sample sent to the network, DQN generates 1 training sample for each action-reward frame. We believe one frame could provide more information than that. For an experience sequence $(..., s_i, a_i, r_i, s_{i+1}, a_{i+1}, r_{i+1}, s_{i+2}, ...)$, we will get two updates for state $s_i$:

$$L_{i,1}(\theta_i) = (r_i + \gamma \max_{a' \in \mathcal{A}^+} Q(s_{i+1}, a'|\theta_i) - Q(s_i, a_i|\theta_i))^2$$

$$L_{i,2}(\theta_i) = (r_i + \gamma r_{i+1} + \gamma^2 \max_{a' \in \mathcal{A}^+} Q(s_{i+2}, a'|\theta_i) - Q(s_i, a_i|\theta_i))^2$$

This update improves the intermediate representation aggressively leading to superior convergence. In practice, we could organize them into one single training vector, as illustrated in the Figure 1.

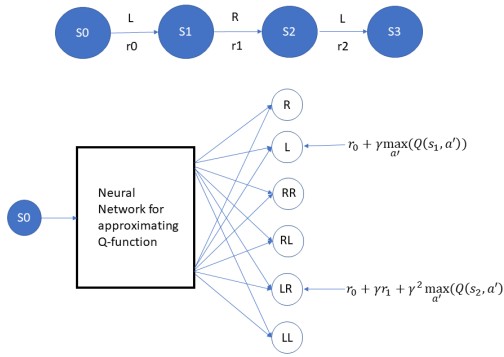

Figure 1: A toy example for ADQN with an enlarged action set $\{L, R, LL, LR, RL, RR\}$. For input $s_0$, we have 2 gradients, one for action $L$ and other for action $LR$.

The same idea can also be applied to the state-of-the-art GA3C (GPU enabled A3C) algorithm (Babaeizadeh et al., 2016). For each state, we choose a meta-action $a^+$ according to the output distribution. If $a^+$ contains only one action, this single action will be executed. If $a^+$ corresponds to an action sequence $(a_1, a_2, ..., a_k)$, these actions will be executed one by one in order.

A4C is a strict generalization of A3C and it allows for three kinds of gradient updates for given action-reward frame: Dependent Updating (DU), Independent Updating (IU), and Switching.

### 2.0.1 DEPENDENT UPDATING(DU)

When we take an action sequence and get rewards, we not only calculate the gradients for this sequence, but also for its corresponding preceding subsequences. For example, in a 2-step multi-action setting, we get an experience queue of $(s_0, a_0, r_0, s_1, a_1, r_1, s_2, ...)$. No matter $(a_0)$ was taken as a basic action or $(a_0, a_1)$ was taken as a multi-step action, we will update both of them for state $s_0$. In this case, we get 2 times more gradient updates as A3C for the same amount of episodes, resulting in aggressive updates which lead to accelerated convergence, especially during the initial phases of the learning.

### 2.0.2 INDEPENDENT UPDATING(IU)

In this scheme, the reward of $a^+$ is the sum of rewards obtained by taking all the basic actions in $a^+$ one by one in order. The next state of $a^+$ is the state after taking all the actions in the sequence. While updating, we only use the information of summed reward, and the next state of $a^+$ without regards to the dependencies and relations between meta-actions.

### 2.0.3 SWITCHING

Our experiments suggest that DU converges faster than GA3C on Atari games for the first few hours of training. Particularly, DU shows a big gap over the speed of original A3C in the Pong game (see Section 3). However, after training for a couple of hours, we observe that aggressive updates cause the network to saturate quickly. This phenomenon is analogous to Stochastic Gradient Descent (SGD)

where initial updates are aggressive but over time we should decay the learning rate (Bottou, 2010). On the other hand, IU offers less aggressive updates and sustains growth in rewards for longer times. To exploit the benefits of both methods, we start with DU and switch over to IU after sometime in the training. For the best outcome, we should switch when DU starts to saturate. Confirming the saturation without human observation is difficult as the rewards are extremely variant. Hence, we set the switching time for each game separately after manually observing the reward curves for DU. Among the games that we report, switching time was 2 hrs for Pong and SpaceInvaders and 2.5 hrs for Qbert and Beamrider. We observe that switching seems to have robust performance in experiments with regards to different choice of hyperparameters.

## 3 EVALUATION ON ATARI GAMES

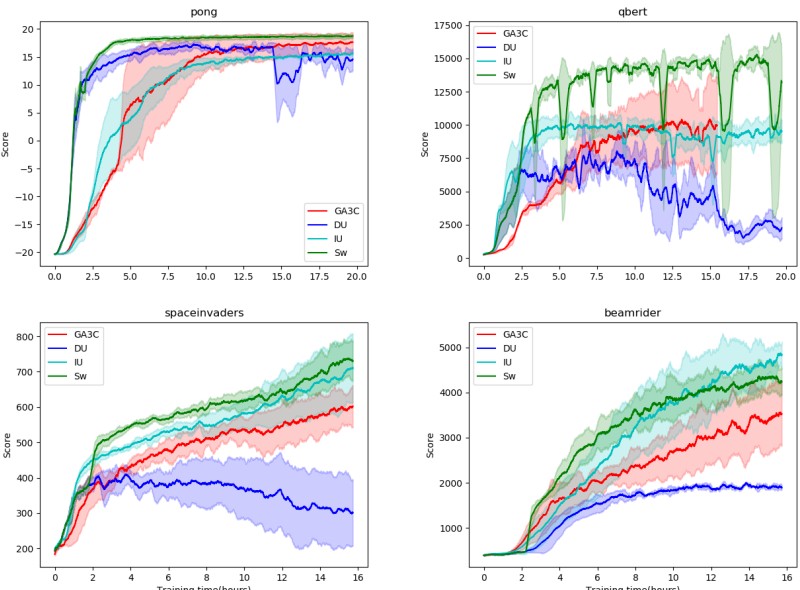

Figure 2: Comparison of three variants of A4C against GA3C. The baseline GA3C is shown in red, Dependent Updates(DU) in blue, Independent Updates(IU) in cyan and the Switching (Sw) in green. The light color fill is one standard deviation away on either side of the mean curve.

Figure 2 demonstrates our A4C experiments on 4 popular and standard Atari-2600 games chosen by GA3C paper. We ran 2-step A4C variants (DU, IU and Switching) and baseline GA3C for each game 3 times and compared the time-wise mean and variance. The machine we used had a 14-core CPU and a single Tesla K20 GPU. The architecture of our network is the same as GA3C paper (32 $4*4$ filters-16 $8*8$ filters-dense layer with 256 nodes-softmax layer with size of enlarged action space).

The plot shows that Switching-A4C consistently beats baseline GA3C by a significant margin on all games. The GA3C scores are very similar to the ones mentioned in original paper. We also infer that DU is suited to games with small action spaces like Pong(3 basic actions) and IU sustains well on larger action spaces like Beamrider(9 basic actions). Switching blends both and is hence the most robust method for varied scenarios.

## 4 CONCLUSION AND FUTURE WORK

We propose a simple yet effective technique of adding anticipatory actions to the state-of-the-art GA3C method for reinforcement learning and achieve significant improvements in convergence and overall scores on several popular Atari-2600 games. We propose a strategy that treats each multi-step action as a sequence of basic actions and extracts gradients for the complete action sequence as well as the preceding sub-sequences. We also identify some issues that challenge the sustainability of our approach and propose simple workarounds to leverage most of the information from higher-order action space. However, the action space grows exponentially with the order of anticipation. Addressing large action space, therefore, remains a pressing concern for future work.

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
