# OpenReview forum: "A4C: Anticipatory Asynchronous Advantage Actor-Critic"
_ICLR.cc/2018/Workshop — Reject_

### Official Review · AnonReviewer2 · 2018-03-02
**A potentially promising RL approach, worth investigating further**

**Rating:** 6
**Confidence:** 4

**Review:**

This paper suggests to extend the action set in A3C with *sequences* of actions, so as to make the model predict the effect of up to k consecutive actions. It is found (on 4 Atari games) that using multiple gradient updates (one for each subsequence of size  <= k in the trajectory) saturates quickly, and better results are obtained by switching during training to a single update associated to the sequence actually  chosen by the agent.

This seems to me like an idea worth investigating, which is why I’m voting for its acceptance. However this is clearly extremely preliminary work and I feel that a lot more needs to be done to better understand its potential and limitations. Including for instance:
- More runs on more Atari games (3 runs on 4 games is clearly not enough)
- Application to DQN (note that Q-learning equations are given but A3C is based on a V critic, not a Q one)
- A much more in-depth investigation of why DU works so poorly (actually worse than IU even at the start on 3/4 games) and why switching to IU makes it improve that much that fast
- A better switching strategy than manual observation
- Trying to take advantage of the structure of sequences of actions, instead of considering them as totally independent: for instance we should have Q(s, ab) <= Q(s, a). Also it seems to me that to behave optimally the agent should only use basic actions (even if predicting the outcome of sequences of actions may be beneficial for training purpose)
- Application to longer sequences than k=2 (may require a different approach to scale)
- Links / comparisons with hierarchical RL techniques

I would also like to ask if in IU the « sum of rewards » is actually the « sum of discounted rewards »? It should be, otherwise longer sequences would benefit from the lack of discount, at the expense of shorter ones.

---

### Official Review · AnonReviewer3 · 2018-03-06
**I liked the paper but it lacks of a proper bibliography**

**Rating:** 7
**Confidence:** 5

**Review:**

This article study the idea of forcing an RL policy to predict/anticipate a sequence of action instead of a single action at each time step. According to the authors the practical benefit of this approach is to speedup learning by getting more gradient from the same number of training episodes.
By applying this multi-step forecasting idea to "Asynchronous Advantage Actor Critic" (A3C), the authors obtain a new algorithm called A4C (the new A comes from "Anticipatory"). Three updating strategies are tested: dependent (some would say "recursive") updating, independent updating and a combination of both.
The experiments on four Atari games corroborates the assumptions that A4C is improving from (G)A3C.
As expected this all-possible-options planning does not scale when the action space is too large.

It was a pleasure to read this (short/workshop/experimental) paper but it definitely calls for a deeper bibliographic work: for instance this notion of "Anticipatory" actions planning was already extensively studied under the name of "options planning" (See "Between MDPs and semi-MDPs: A framework for temporal abstraction in reinforcement learning" by Sutton et al.). In the forecasting literature the key words are "multi-step forecast".

Minor remark: the experiments on Atari games are great, but a few well-chosen experiments on simple MDPs could provide some insights on this anticipatory/options behavior.

---

### Official Review · AnonReviewer1 · 2018-03-09
**A mix of n-step returns and options ends in unclear experimental results.**

**Rating:** 3
**Confidence:** 4

**Review:**

This paper seems to present a version of n-step returns along with predefined options that are simple series of base actions.  There is no mention of n-step returns, options, or the bias-variance tradeoff of using n-step returns vs. 1-step returns.  Additionally, the authors claim to have better performance than A3C, but chose to switch between an n-step return and a 1-step return at a specific point in training, that is chosen once a training run has already been performed in entirety against a certain environment.  This means that to train a policy using the proposed algorithm, one needs to perform one full (or almost full) run, subjectively choose a switching point, and then re-run training form scratch.  Given this fact, and that the authors don't position any of their work relative to options & n-step returns, I don't believe this paper is sufficiently thorough to warrant presenting at the ICLR 2018 workshop track.

---

> ### Comment · AnonReviewer1 · 2018-03-09
> **Followup**
>
> If you would like to continue advancing in this direction, please have a look at chapters 7, 12 & 17.2 in Rich Sutton's 2nd edition RL book: http://incompleteideas.net/book/bookdraft2018feb28.pdf

---

### Public Comment · ~Oriol_Vinyals1 · 2018-02-17
**Please Fix Length**

Your paper violates by a few lines the 3 page limit (see https://iclr.cc/Conferences/2018/CallForWorkshops). Please send us a fixed version of your PDF at iclr2018.programchairs@gmail.com by the end of Monday, February 19th, or else we will reject your paper.

Thanks,
ICLR2018 Program Chairs

---

> ### Public Comment · ~Tharun_Medini1 · 2018-02-18
> **I've sent the updated file**
>
> I've emailed the fixed paper with the subject 'A4C - Corrected version of our workshop paper' to iclr2018.programchairs@gmail.com. Thank you for the consideration.

---

### Decision · Program_Chairs · 2018-03-20
**ICLR 2018 Workshop Acceptance Decision**

**Decision:**

Reject

**Comment:**

Based on the reviews, this paper has not been accepted for presentation at the ICLR workshop. However, the conversation and updates can continue to appear here on OpenReview.